# DEM Analysis of Track Ballast for Track Ballast–Wheel Interaction Simulation

**Nam-Hyoung Lim** [1], **Kyoung-Ju Kim** [2], **Hyun-Ung Bae** [2] **and Seungjun Kim** [3,*]

1    Department of Civil Engineering, Chungnam National University, Daejeon 34134, Korea
2    Road Kinematics Co., Ltd., Cheonan 31094, Korea
3    School of Civil, Environmental, and Architectural Engineering, Korea University, Seoul 02841, Korea
*    Correspondence: rocksmell@korea.ac.kr; Tel.: +82-2-3290-4868

**Abstract:** This study aims to suggest a rational analysis method for a track ballast–wheel interaction that could be further developed to model the interaction in a train-derailment event, based on the discrete-element method (DEM). Track ballast is filled with gravel to form the trackbed. Although finite-element analysis (FEA) is widely applied in structural analysis, track ballast cannot be analyzed using conventional FEA because this approach does not allow separation of elements that share nodes. The DEM has been developed to analyze the dynamic behavior of separable objects, assuming that the objects are rigid. Therefore, track ballast can be modeled as separable rigid pieces of gravel, and its dynamic behavior can be analyzed using a rational contact model. In this study, a rational numerical strategy for track ballast–wheel interaction was investigated using the DEM approach. The suggested analysis method was validated through comparison with the experimental results of a drop test. In addition, case studies were conducted to investigate the effects of the contact-model parameters on the simulation result.

**Keywords:** track ballast; gravel; DEM; explicit dynamics; contact model; drop test

## 1. Introduction

Derailment of a rapidly moving train may lead to fatal consequences, thus such accidents should be prevented through effective means, such as derailment-prevention devices and structures. In Europe and in Japan, Korea, and several other countries, effective derailment-prevention methodologies have been studied and developed, such as derailment-containment walls [1–4]. In addition to the fundamental prevention concepts, these studies have suggested design methods and provisions for the walls. Recently, a new type of structural system for derailment-containment prevention (DCP), which is constructed between the rails, has been developed in Korea [4], as shown in Figure 1. The concrete structural member is located between rails to restrict the horizontal movement of derailed train. To design the structural member and to validate its performance as a DCP system, the structural responses of the member under contact, and the impact forces induced by a derailed train, should be investigated by rational procedures.

Through simulations and experiments, the performance of the newly proposed system has been verified, and an appropriate design procedure for this system has been suggested [1,4]. However, the prevention performance has been investigated only for concrete track ballast, even though in several cases, the track ballast can be constructed as a gravel track bed. This indicates that, apart from concrete ballast, the DCP systems should be additionally examined for train derailment over a gravel ballast.



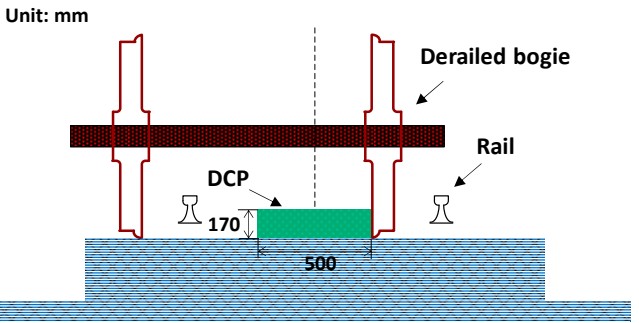

**Figure 1.** Overview of the derailment-containment prevention (DCP) wall developed in Korea [1].

As mentioned before, in several cases, the track ballast comprises pieces of gravel of irregular shapes, which form the track bed, as shown in Figure 2. When a train is derailed, the wheel of the derailed train will contact the ballast, and the interaction between the gravel pieces and the contacted wheel will affect the subsequent behavior of the derailed train. Therefore, for designing, verifying, and optimizing rational DCP systems with gravel ballast systems, this interaction should be analyzed to investigate the behavior of the derailed train after it contacts the ballast.

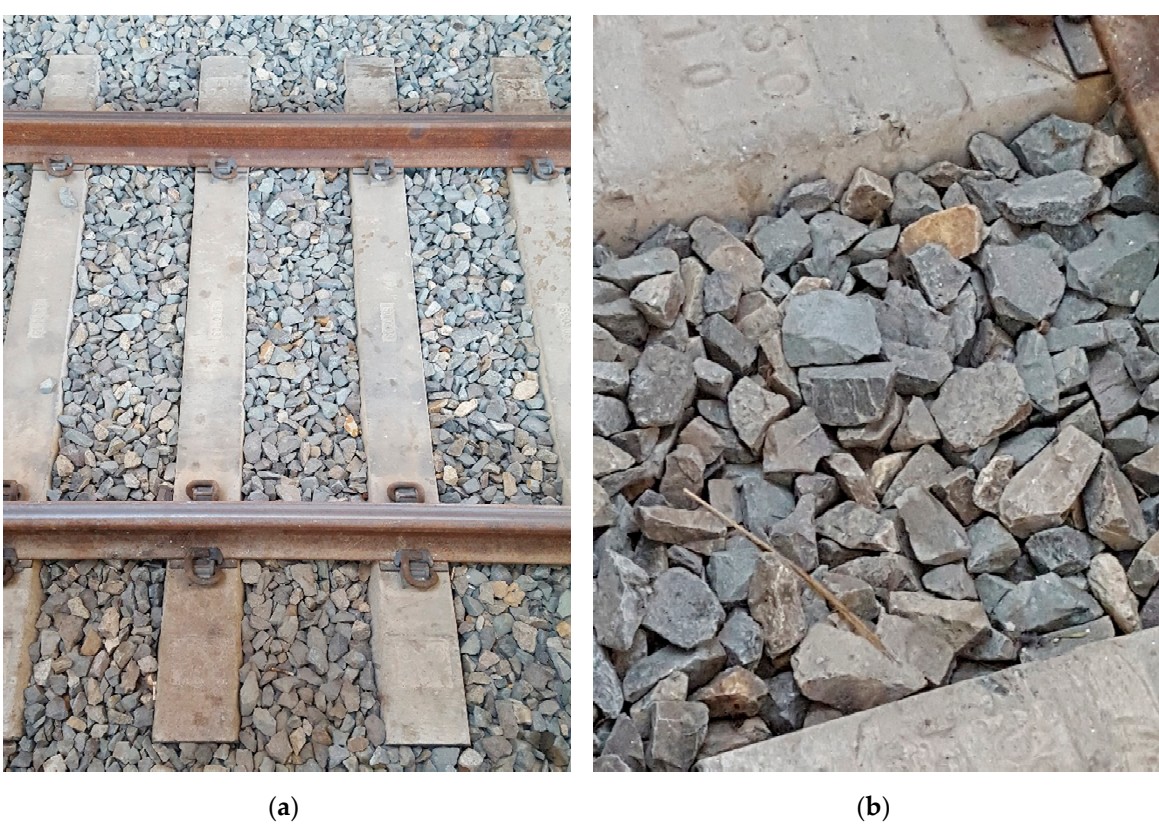

(**a**)    (**b**)

**Figure 2.** Conventional track ballast (ballast for rapid trains in Korea): (**a**) track ballast; (**b**) pieces of gravel for the ballast.

For the modeling and simulation of the track ballast, the finite-element method (FEM) has been widely used because of its reliability and convenience. By using the conventional FEM approach, the ballast is modeled as shell or solid elements, with material properties equivalent to those of gravel layers in the two-or three-dimensional domain, respectively. Even when appropriate material models are applied, there is a limitation regarding the interaction between the gravel pieces in contact. The gravel pieces can be in contact or separate from each other. However, if the gravel layer is

modeled using the conventional FEM, the interaction characteristics cannot be effectively considered because this approach does not allow the separation of elements that share nodes and exhibit a very large deformation. To rationally design the DCP, the post-behavior of the derailed train should be analyzed first to evaluate the required physical quantities, including the impact load on the DCP structural members due to the derailed train. When the train is derailed over gravel ballast, it can be assumed that the interaction between the ballast and train, which is induced by the contact between the gravel and the wheel of the bogie, directly affects the subsequent behavior of the derailed train. Therefore, rational numerical modeling and simulation methods for the ballast should be studied to perform the post-behavioral analysis of a train derailed over gravel ballast.

As mentioned earlier, the track ballast has been modeled based on FEM approaches. Paderno [5] studied the long-term settlement characteristics and the effect of the tamping process on the dynamic behavior of the ballast, using FEM approaches. In that study, the ballast was modeled as shell elements, and the equivalent material properties, such as internal friction angle, dilation angle, and cohesion, which had been evaluated through a simple experiment, were applied using the Mohr–Coulomb plastic model. Although this approach could be applied in ballast modeling for relatively small-deformation problems, it still has limitations with respect to directly considering the interactions between the pieces of gravel in contact, especially the interlocking effect between these pieces. Zhou et al. [6] studied methodologies for analyzing the ultimate lateral pressure of penetrated pile in undrained clay based on FEM. Although the approach can be applied for relatively large deformable soil layers, separation of the gravels cannot be simulated by the approach.

Ahmadi and Eskandari [7,8] suggested the vibration analysis method of a rigid circular disk embedded in a transversely isotropic solid. Eskandrai et al. [9] studied the closed-form solution for lateral translation of an inextensible circular membrane embedded in a transversely isotropic half-space. However, there are still limitations for considering the time-varying change of the geotechnical properties of the ballast due to the applied forces.

To overcome these limitations, other approaches have been studied based on the discrete-element method (DEM). Using DEM, each piece of gravel can be modeled as an individual object with efficient calculation; therefore, the interaction between the gravel elements in the ballast, including slip, separation, and re-attachment, can be analyzed. Zhou et al. [10] performed a DEM-based analytical study to simulate the effect of the tamping process on the compactness of track ballast. Mahmoud et al. [11] studied a simulation method for the permanent settlement of track ballast due to cyclic loading, based on the DEM approach. Kim et al. [12] investigated the influencing factors in ballast settlement using a DEM simulation. Furthermore, Pi et al. [13] studied the relation between the geogrid rib size and the particle size distribution of ballast materials using a DEM simulation.

When DEM approaches are used, the modeling method used for the gravel, including the shape and initial positions of the gravel pieces, is important. Thakur et al. [14] studied a modeling method for arbitrarily shaped gravel pieces using a clump of circles in 2D for DEM simulation. Mollon and Zaho [15] studied the generation and particle-packing method of sand in a DEM simulation. They suggested a method to determine the number and initial positions of the particles to satisfy the pre-defined size distribution, shape, and density of the sand layer. Campello and Cassares [16] also studied a method to rapidly generate a particle layer using the DEM approach. In recent studies, the individual particles have been modeled as sphere-type rigid bodies or clumps consisting of spheres, to ensure efficient calculation. Nezami et al. [17] adopted a DEM simulation approach using specifically shaped objects with fast tracking for contact between the objects, using their in-house simulation code, DBLOCK3D. As alternatives to the conventional FEM, DEM approaches are being used to model and simulate ballast. If a rational modeling and simulation method for track ballast is used, the simulation method for analyzing the behavior of the derailed train can be investigated considering the interaction between the ballast and the derailed bogie. For the simulation, DEM approaches can be effectively used.

This study aims to propose a method for modeling track ballast filled with gravel pieces and simulating a ballast–wheel collision. Because the pieces of gravel can be separated and re-attached,

conventional FE methods, which do not allow the separation of the elements, exhibit limitations regarding the modeling of gravel-filled ballast. To overcome this limitation, DEM-based modeling and simulation methods were investigated in this study. Using a DEM approach, the individual pieces of gravel can be modeled; hence, the interactions between the pieces of gravel in contact and between the gravel and other objects can be simulated. For a rational simulation of the ballast–wheel interaction, appropriate modeling and simulation strategies should be adopted considering the characteristics of the gravel used for the track ballast. To consider the interlocking of the gravel elements in contact, a clump-type rigid object is mainly used for each piece of gravel. Whereas the gravel pieces are modeled based on DEM, the other objects that contact and clash with the gravel layer are modeled based on FEM to calculate and consider the energy absorption of the deformable bodies. Thus, a DEM–FEM combined simulation method is adopted in this research.

An experiment was conducted to examine the behavior of a freely dropped object that clashes with the gravel layer. To validate the simulation method, the motions of the dropped weight, obtained using both the experiment and the simulation, were directly compared. In addition, case studies were conducted to investigate the effects of the parameters of the gravel–gravel and gravel–object contact models on the simulation results for the ballast–wheel interaction.

## 2. Theoretical Background for Ballast Modeling and Analysis

Several researchers have modeled ballast using the FEM with shell and solid elements for two- and three-dimensional analyses, respectively. In such studies, the equivalent soil properties, such as elastic modulus, Poisson's ratio, internal friction angle, dilation angle, and cohesion, should be defined using appropriate material models such as the Mohr–Coulomb model. Thus, the equivalent material properties should be evaluated for modeling track ballast filled with irregular pieces of gravel.

Although the FEM has been widely used in numerical simulations of various structural and geotechnical engineering problems, it has significant limitations regarding the modeling and simulation of the behaviors of a track ballast filled with gravel, because of the fundamental assumptions of the method. Using the FEM approach, the interlocking and separation of the gravel pieces cannot be directly considered because the conventional FEM does not allow the separation of elements attached to each other. Specifically, the collision between the ballast and the wheel of a derailed train cannot be rationally simulated using conventional FE approaches because of this limitation. Thus, the ballast should be modeled using another approach.

The DEM is one of the alternative solutions for modeling track ballast. Using the DEM approach, each piece of gravel can be modeled as an individual object that does not share nodes. Thus, all the pieces of gravel in contact can be detached and re-attached after their positions are updated. Therefore, the limitation of the FE approach can be overcome using the DEM. Based on the advantages of the DEM approach, an appropriate simulation strategy for modeling gravel ballast and simulating ballast–wheel collisions is suggested in this study.

### 2.1. DEM Analysis

In a DEM analysis, the individual objects are modeled as rigid bodies. The simplest way to model gravel is to use spherical rigid bodies. For modeling gravel pieces as simple rigid bodies, an appropriate contact model should be used to consider the elasticity of the actual object. Hence, after modeling the rigid bodies using the contact model between them, the analysis can be conducted. The analysis is performed based on Newton's second law. Hence, the increments in the velocity and displacement of the individual bodies due to the applied loads are determined by calculating their acceleration. Figure 3 shows the general procedure of the DEM analysis based on an explicit structural dynamic analysis.

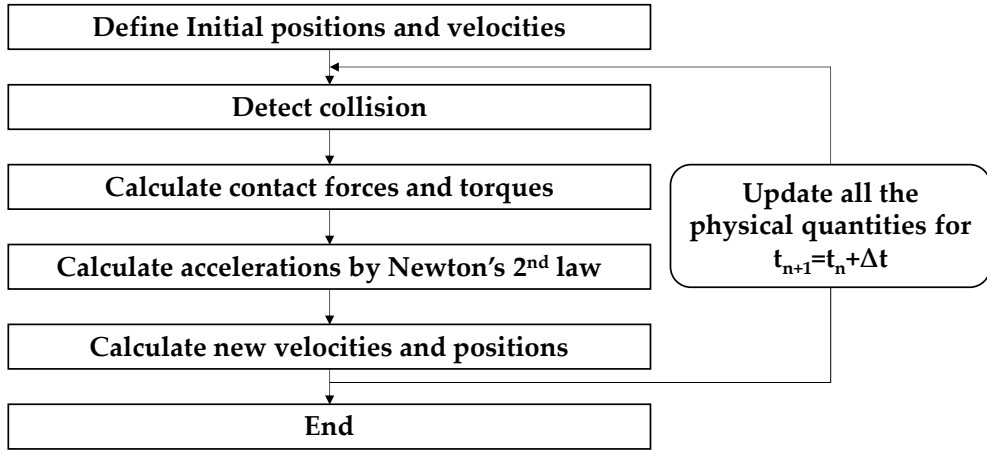

**Figure 3.** General procedure of discrete-element (DEM) analysis based on explicit structural dynamic analysis.

At the preparation stage, the initial positions and velocities of the rigid bodies are defined. Because all the objects are considered as individual rigid bodies, initial overlapping between contacted objects is not allowed. Therefore, if the initial position of each object is defined, the initial analysis should be conducted to determine the appropriate position without any overlapping. To avoid the initial analysis and define the appropriate initial positions of the objects, a numerical method such as particle packing can be used. The details of particle packing are described in the next section.

Once the initial conditions are defined, including the initial positions, velocities, and boundary conditions of the domain, the dynamic analysis can be conducted to obtain the dynamic structural responses of all the objects, including the rigid bodies and deformable structural members, in the model. Again, the DEM analysis is suitable for dynamic problems because the displacement increments are calculated by integrating the acceleration increments. As shown in Figure 2, the collision between individual bodies is verified to determine whether the contact force and torque should be calculated or not. If the force and torque of the object are not calculated, the acceleration increment of the object is calculated using the force and moment determined at the previous time step. If the force and torque have to be calculated, the forces are calculated based on the pre-defined contact model. Various contact models exist for DEM analysis. Among these, the Hertz-based model was used in this study because the model can be effectively applied for considering contact behaviors of elastic bodies modelled as rigid spherical bodies. The applied contact model is shown in Figure 4.

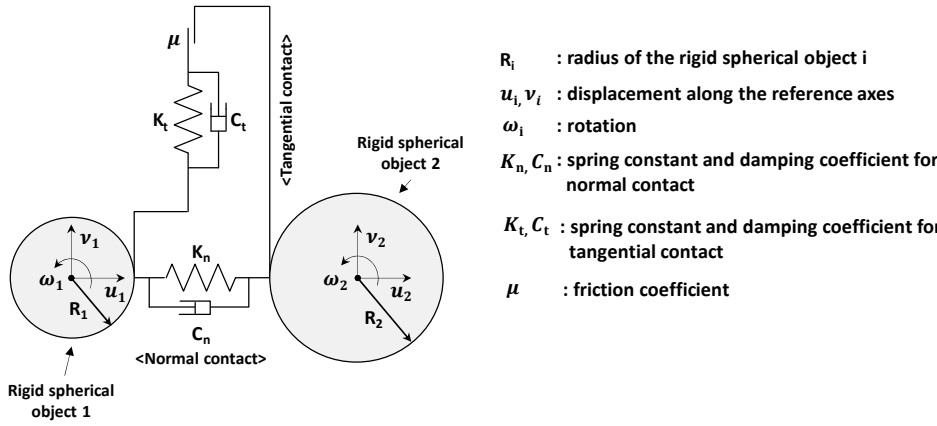

**Figure 4.** Applied contact model for considering contact between the rigid particles.

Using the contact model, the elastic behavior of the object can be taken into account, even though the objects are modeled as rigid bodies for numerical simplicity. Thus, the stiffness for normal contact is estimated considering the elastic modulus and the size of the objects in contact. In addition, the friction between the objects in contact is considered as a tangential contact force. Once the friction coefficient for the objects in contact is defined, the friction force can be calculated because the normal force has been calculated. In this simulation, a simple linear friction model was applied. Consequently, all the force components required for defining the motion equation of each object, including inertia and damping forces, were calculated and considered. The normal contact force $F_n$ between two spherical particles in contact can be calculated using the Hertz solution, as shown in Equation (1):

$$F_n = \frac{4}{3}E^* \sqrt{R} \sqrt{\delta^3},$$
(1)

where $R = \frac{R_1 R_2}{R_1 + R_2}$; $\frac{1}{E^*} = \frac{1-v_1^2}{E_1} + \frac{1-v_2^2}{E_2}$; $E_i$ is the Young's modulus of particle $i$; $v_i$ is the Poisson's ratio of particle $i$; and $\delta$ is the normal overlap between two spherical particles in contact.

As shown in Equation (1), evaluation of $\delta$ is very important because it determines the contact force $F_n$. In the simulation, $\delta$ was evaluated as the approach distance between remote points on the contacting spheres in every time increment during explicit dynamic analysis.

The tangent contact force $F_t$ can be calculated using the pre-defined friction coefficient μ, while the damping forces $F_n^D$ and $F_t^D$ can be calculated using the pre-defined damping coefficients $C_n$ and $C_t$, respectively.

As mentioned previously, a rational definition of the initial position of the objects is important. Particle packing is one of the useful methods for modeling a layer filled with rigid particles. In this study, a random particle-packing method was primarily used to model the track ballast filled with irregularly sized gravel pieces. Figure 5 presents the general procedure of the packing method.

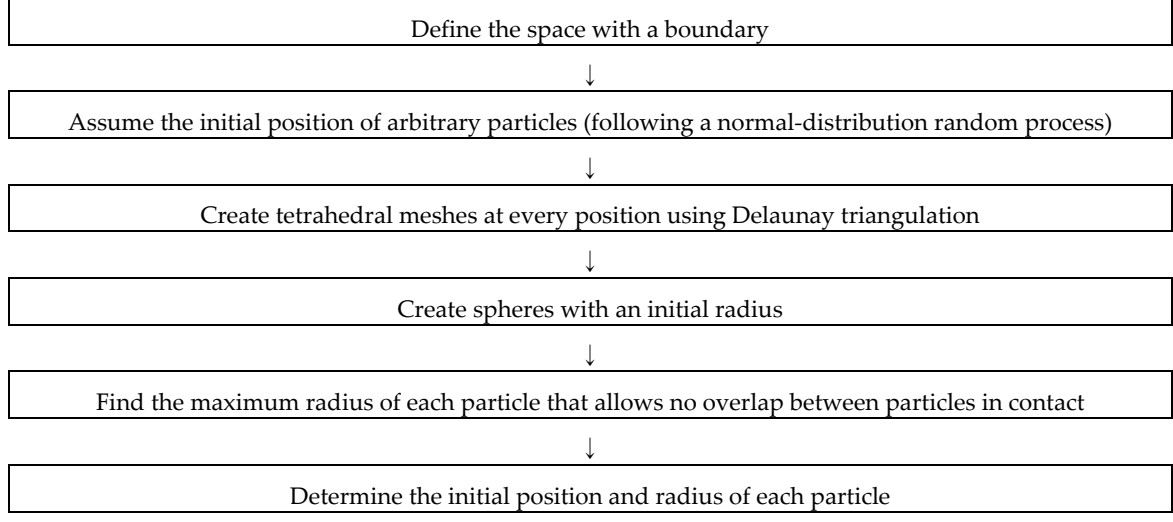

**Figure 5.** General procedure for close packing of particles.

*2.2. Korean Regulations for Size Distribution of Ballast Gravel*

In Korea, the size distribution of the gravel for track ballast is determined according to the specifications for railroad equipment [18]. Table 1 shows the details of the size distribution of the gravel for track ballast presented by the regulation.

**Table 1.** Size distribution of the gravel for track ballast used in Korea [18].

| Size (mm) | 10 | 22.4 | 31.5 | 40 | 50 | 63 |
|---|---|---|---|---|---|---|
| % passed by sieve analysis | - | 0–5 | 5–35 | 30–65 | 60–100 | 100 |

The procedure shown in Figure 5 is followed using the given number of particles. Thus, this procedure can be performed to determine the maximum radius of each particle for the given boundary with the given number of particles. To achieve the required size distribution for the gravel layer, the procedure shown in Figure 5 should be revised. First, the number of particles should be treated as one of the variables to be determined via the optimization process. Therefore, the variables to be found are the number of particles and the size and initial location of each particle. The boundary is considered as the dimensions of the target layer to be formed. The object functions to be minimized are defined by calculating the overlapping length and void of the layer. Figure 6 shows the revised process for the close packing.

The revised procedure can be successfully performed based on the genetic algorithm, which is a powerful optimization algorithm. Using this procedure, close packing of the particles can be obtained considering the given size distribution.

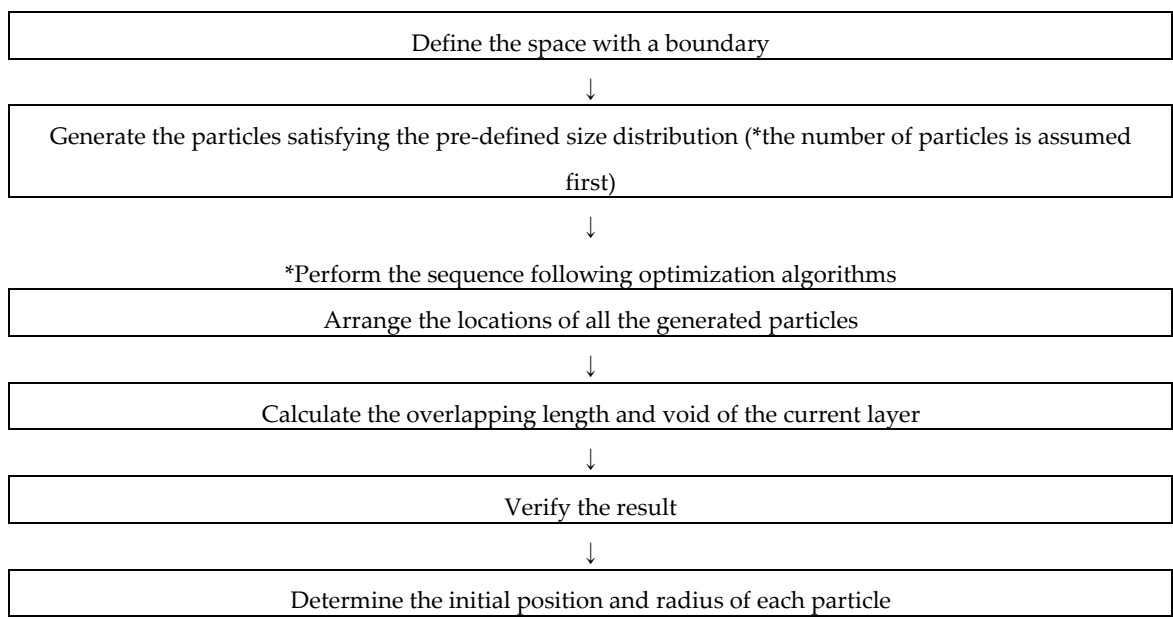

**Figure 6.** Revised general procedure for close packing of the particles.

## 3. Strategy and Validation of Numerical Implementation

### 3.1. Simulation Strategy for Ballast–Wheel Collision

Ballast–wheel collisions were analyzed to trace the path of a derailed train and investigate the effects of the ballast on the energy absorption under various conditions, including the ballast condition and kinetic characteristics of the derailed train. To investigate the behaviors, the change in the position, forces, and energies of the clashing objects should be simulated in the time domain. Therefore, a DEM–FEM combined simulation should be conducted, wherein the ballast is modeled using a DEM approach, and the colliding object is modeled using a FEM approach. Figure 7 shows the procedure for the simulation of the ballast–wheel collision.

| Model the ballast filled with gravel pieces considering the given size distribution (using DEM) |
| :---: |

↓

| Model the object that will clash with the ballast (using FEM) |
| :---: |

↓

| Define the analysis protocol, including initial time step, total time, load sequence, and initial and boundary conditions |
| :---: |

| Define the contact model between the gravel pieces and gravel–wheel |
| :---: |

↓

| Conduct an explicit dynamic analysis to calculate the motion, force, and energies of all the objects in the time domain |
| :---: |

**Figure 7.** Simulation procedure for ballast–wheel collision.

### 3.2. Validation

To validate the numerical strategy, a drop test was conducted. As shown in Figure 8, a 1000 × 1300 × 700 mm (width × breadth × height) steel box was filled with gravel pieces up to a height of 600.0 mm. The sizes of all the gravel pieces were 31.5–50.0 mm, which is suitable for track ballast. The 656-kg weight was freely dropped from heights of 0.5 and 1.0 m. During the drop test, the motion of the weight was measured in the time domain. Furthermore, the depth of penetration into the gravel layer was measured. To prevent a local effect due to deformation of the box, the wall of the box was sufficiently stiffened. The test was conducted twice for each drop height. During the test, the acceleration of the weight was measured via vision-postprocessing in the time domain, and then, the velocity and displacement were evaluated via numerical integration of the measured acceleration.

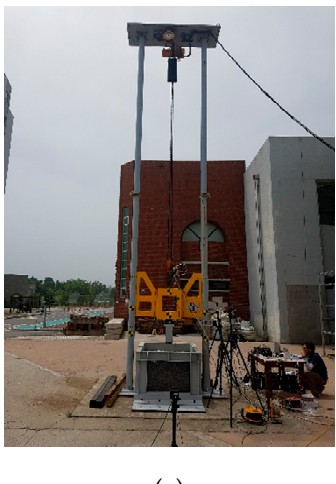

(**a**)

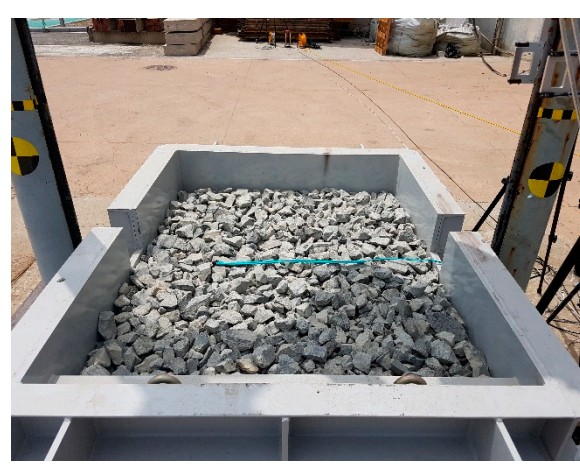

(**b**)

**Figure 8.** *Cont.*

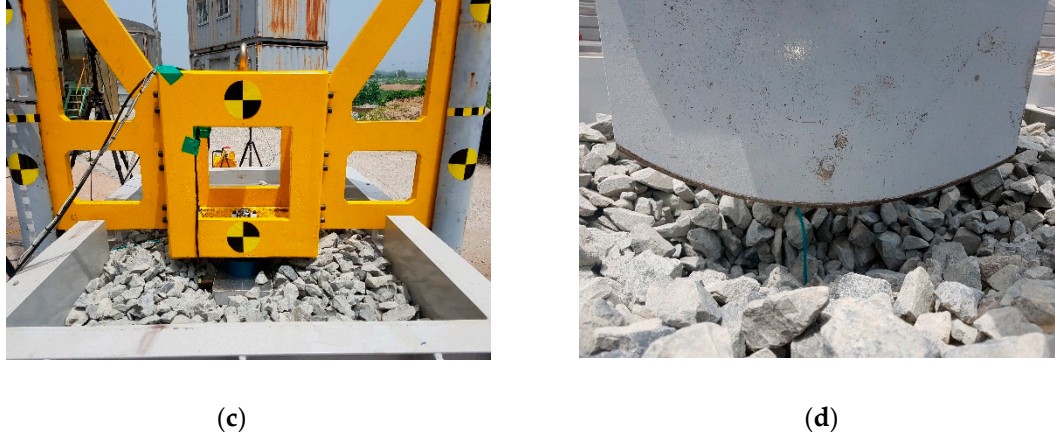

(**c**)                                                                 (**d**)

**Figure 8.** Drop test to simulate the collision between the dropped weight and gravel layer: (**a**) Front view of the drop test; (**b**) gravel layer; (**c**) weight penetration; (**d**) gravel layer after penetration.

Figure 9 shows the numerical model for simulating the drop-test with the same geometric conditions. As shown in the figure, the gravel layer was modeled based on DEM, while the weight was modeled using FEM. In this simulation, the gravel was modeled as simple rigid sphere clumps. Thus, there was a limitation regarding the interlocking effect between the arbitrarily shaped pieces of gravel. This could be one of the sources of the difference between the results from the experiment and the test. Following the procedure shown in Figure 7, explicit dynamic analyses were conducted, and the motion of the dropped weight was obtained. In addition, after the simulation, the penetration depth and the motion of the dropped weight were observed for comparison with the result from the drop test. The DEM–FEM simulation was conducted using ABAQUS V6.17 [19]. The particle packing was performed using a MATLAB-based in-house code, following the procedure shown in Figure 5. After modeling, the explicit dynamic analyses were conducted. The initial time step was set as 1.0 E−5 s, and the total simulation time was 1.5 s. After simulation of the drop test, the vertical motion and final penetration depth of the weight were evaluated and compared with those obtained from the experiment.

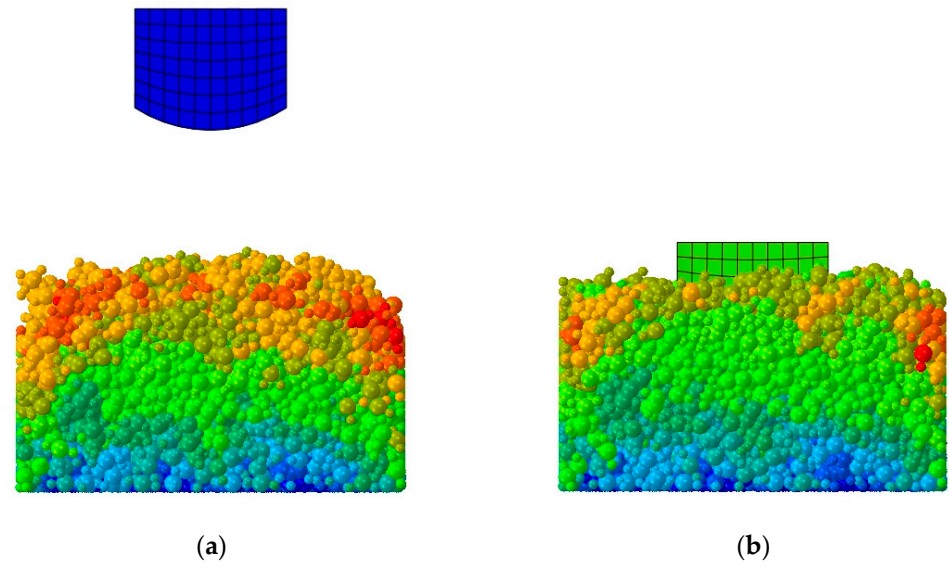

(**a**)                                                                 (**b**)

**Figure 9.** *Cont.*

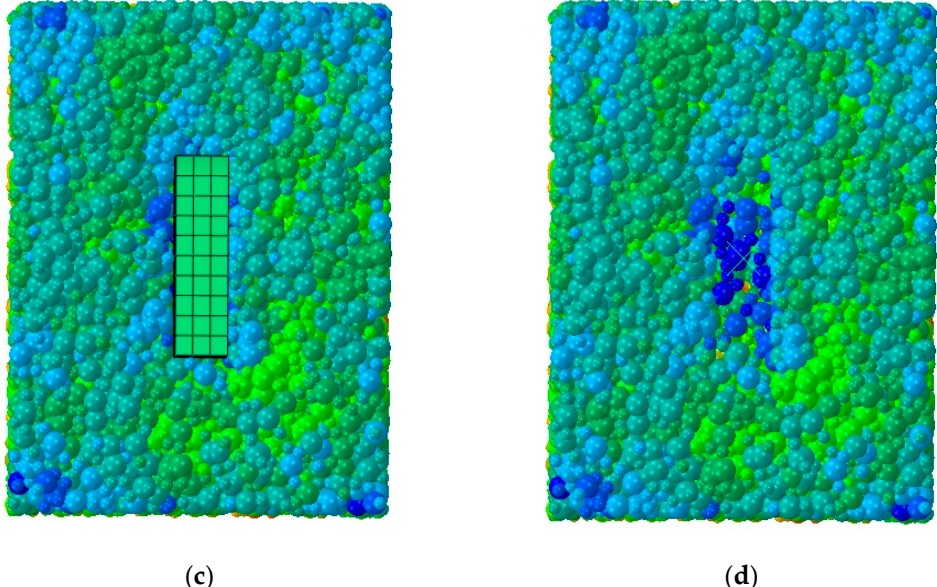

(**c**)                                               (**d**)

**Figure 9.** DEM–FEM simulation of the drop tests: (**a**) dropping the weight; (**b**) penetration after collision; (**c**) weight stopped; (**d**) interrupted gravel layer due to penetration.

Figures 10 and 11 show the comparison between the motions of the weights, obtained from the drop test and the simulation. In the simulation, different friction coefficients for the gravel pieces in contact were applied to verify the effect of friction between the gravel pieces. Although the curves are not perfectly fitted, the suggested method can be adopted for the simulation of ballast–wheel interactions because the tendencies of the motion of the weights are similar. From the curves, we can select the time for the first contact between the dropped weight and the gravel layer, following which, the velocities and changes in displacement of the weight can be directly compared. As shown in the figures, the curves of the velocity and displacement of the weights obtained from the tests and the simulation are similar, including the magnitudes and the time required for full penetration.

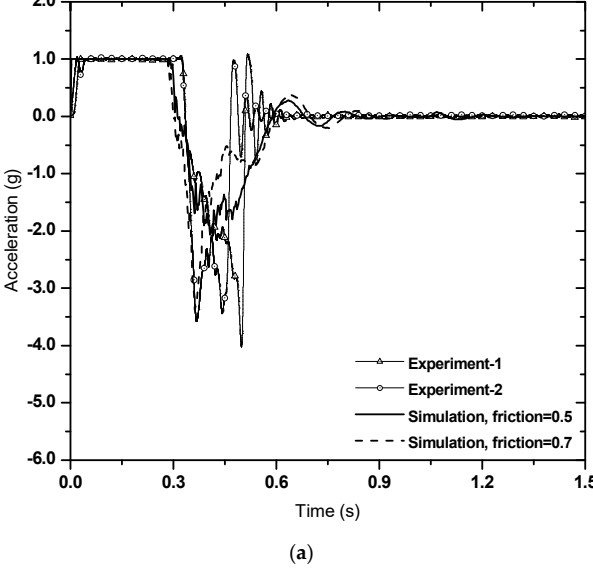

(**a**)

**Figure 10.** *Cont.*

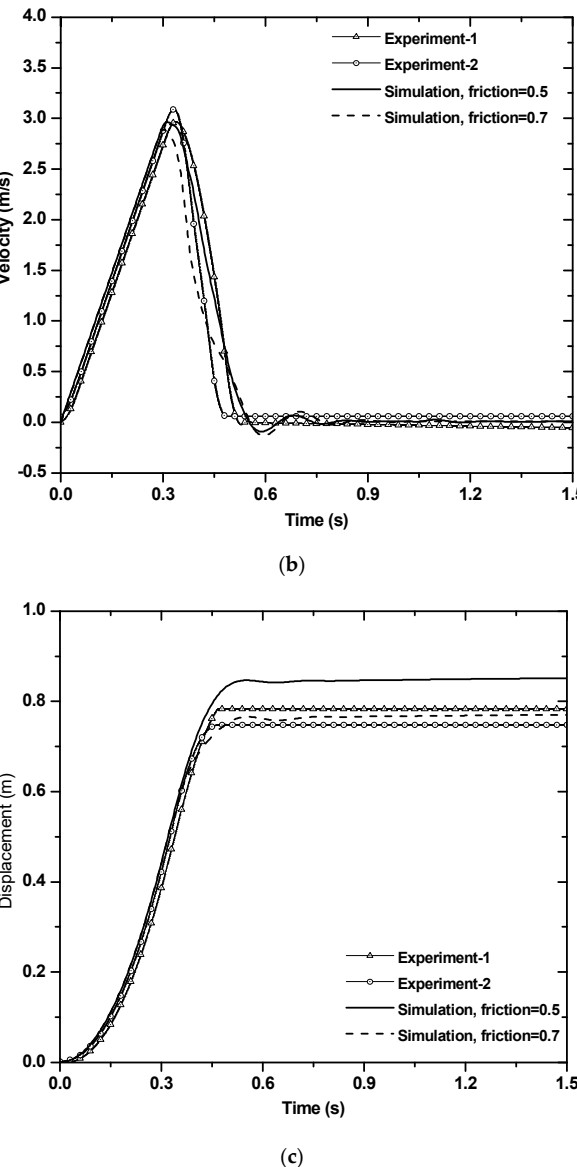

**Figure 10.** Comparison between the motions of the dropped weight, obtained from tests and simulations (dropped from 0.5 m height): (**a**) acceleration; (**b**) velocity; (**c**) displacement.

The displacement curves obtained from the postprocessing of the test, shown in Figure 9c, reveal significant rebounding after full penetration. However, this is not the actual situation; it originated from the cumulative error of the numerical integration. To verify further, the penetration depth after the test was compared with that after the simulation.

As shown in Table 2, the penetration depths measured from the tests and simulations were very similar. The simulation results clearly show that the penetration depth is affected by the applied friction coefficient between the pieces of gravel. Friction is one of the influencing factors in the interlocking between the pieces of gravel in contact. It can be expected that large friction induces extensive interlocking between the gravels. The physical mechanism is clearly expressed in the simulation results. The test results are not well-matched with each other because tamping was not performed after creating the layer. If tamping had been conducted, the layers would have had very similar properties.

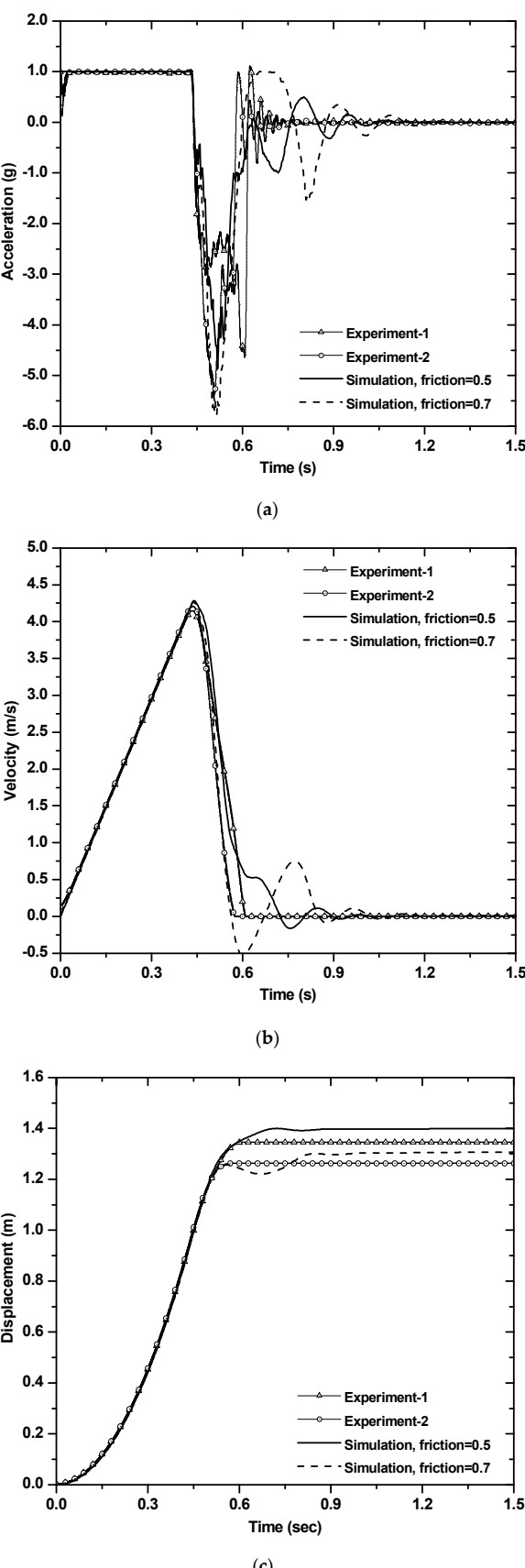

**Figure 11.** Comparison between the motions of the dropped weight, obtained from tests and simulations (dropped from 1.0 m height): (**a**) acceleration; (**b**) velocity; (**c**) displacement.

**Table 2.** Comparison between penetration depths.

|                   | Test   |        |         | Simulation     |                |
|-------------------|--------|--------|---------|----------------|----------------|
|                   | Test-1 | Test-2 | Average | Friction = 0.5 | Friction = 0.7 |
| 0.5-m drop height | 0.33   | 0.24   | 0.28    | 0.35           | 0.26           |
| 1.0-m drop height | 0.46   | 0.35   | 0.41    | 0.47           | 0.38           |

## 4. Case Study: Effects of Contact Model Parameters on Simulation Results

### 4.1. Effect of Friction between Gravel Pieces

In this section, the effect of the friction between the pieces of gravel on the simulation results is described in detail. In the case study, a friction coefficient of 0.4–0.7 was applied in the contact model of the gravel and, then, the motion of the penetrating weight and the friction-energy absorption were numerically compared to investigate the interlocking due to the applied friction coefficients.

As shown in Figure 12, the friction between the pieces of gravel significantly affects the motion of the weight after the impact. The equivalent geotechnical properties of the gravel layer are determined by the effect of interlocking between the contacted gravel pieces. Based on the simulation results, it can be concluded that the considered friction coefficient between the pieces of gravel governs the geotechnical properties of the gravel ballast, including the rigidity of the layer.

Figure 13 shows the energy dissipation due to friction in the different gravel layers with different friction coefficients between the gravel pieces. As shown in the figure, the energy dissipation due to friction decreases as the friction coefficient increases. This clearly proves that the layer would be stiffened owing to the considerable interlocking effect induced when a larger friction coefficient between the pieces of gravel is applied. Thus, the friction coefficient between the gravel pieces is one of the most influential factors for the simulation.

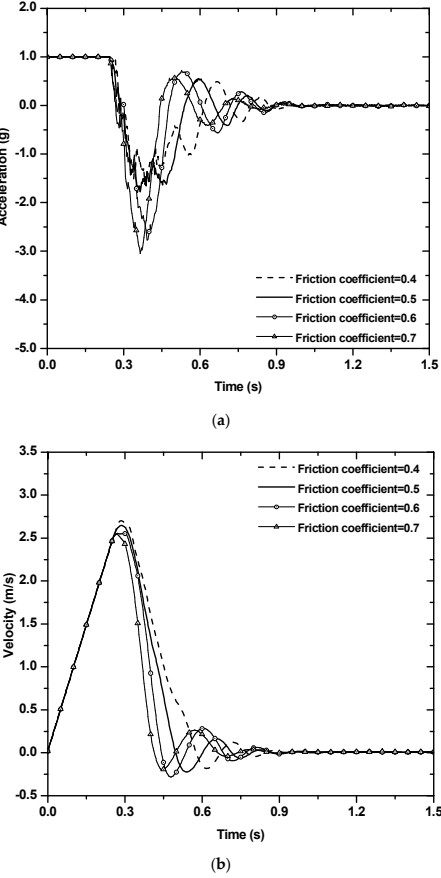

**Figure 12.** *Cont.*

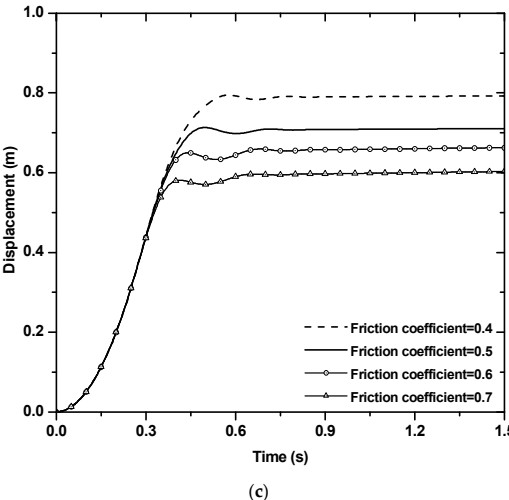

(**c**)

**Figure 12.** Vertical motion of weights clashing with gravel layers for different friction coefficients between the gravel pieces (friction coefficient between a gravel piece and the weight = 0.6): (**a**) acceleration; (**b**) velocity; (**c**) displacement.

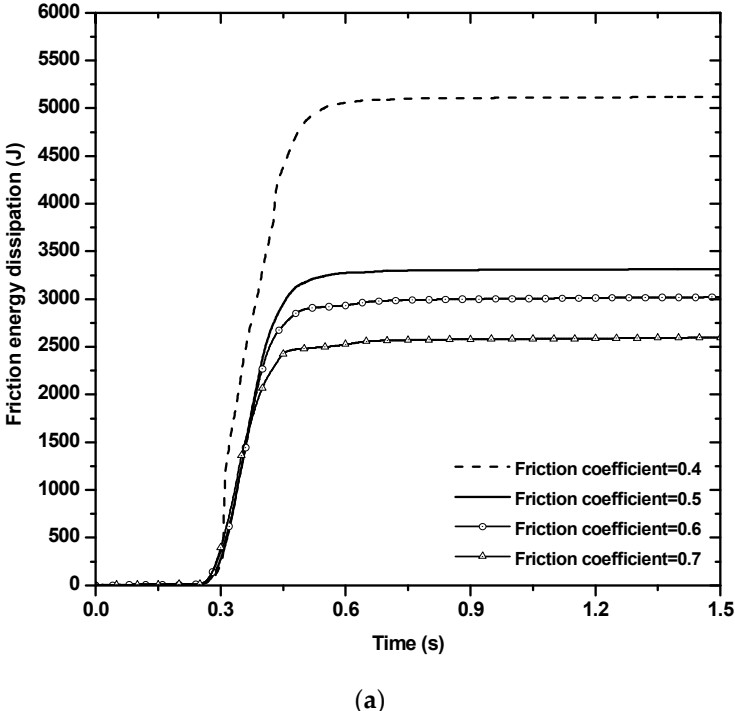

(**a**)

**Figure 13.** *Cont.*

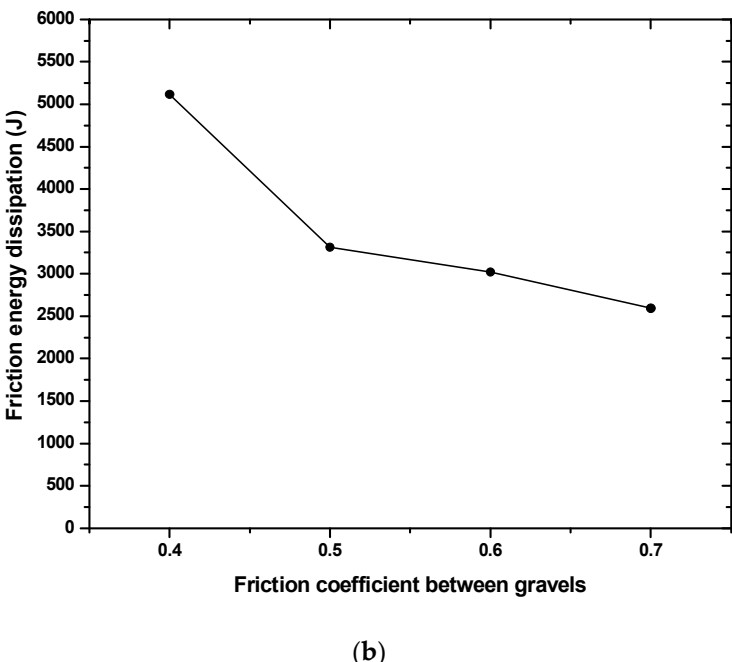

(**b**)

**Figure 13.** Comparison of friction energy dissipation with different friction coefficients between pieces of gravel: (**a**) variation; (**b**) change of maximum friction dissipation owing to the considered friction coefficient.

## 4.2. Effect of Friction between Gravel and Dropped Weight

In addition to the gravel pieces in contact, the friction between the gravel and the object may be one of the influential factors for the simulation result. To investigate the effect of this factor on the result, a case study considering various friction coefficients between the gravel and the weight was conducted. For the parametric study, a friction coefficient of 0.4–0.7 was applied to the DEM–FEM model, with a friction coefficient of 0.5 between the gravel pieces. Figure 14 shows the vertical motion of the weights dropped from a height of 0.5 m.

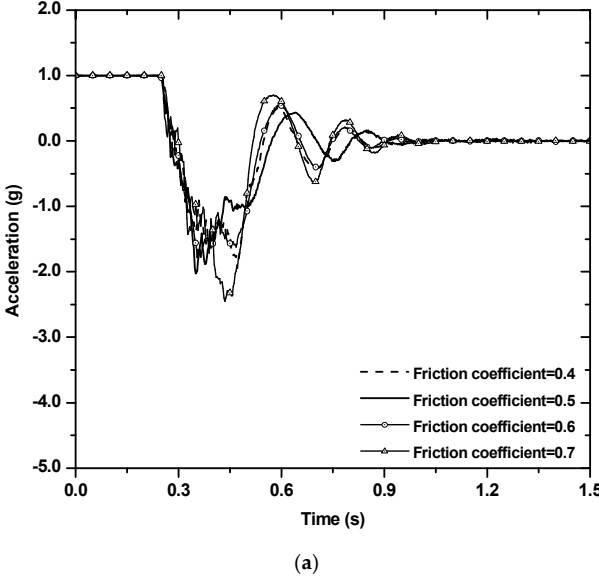

(**a**)

**Figure 14.** *Cont.*

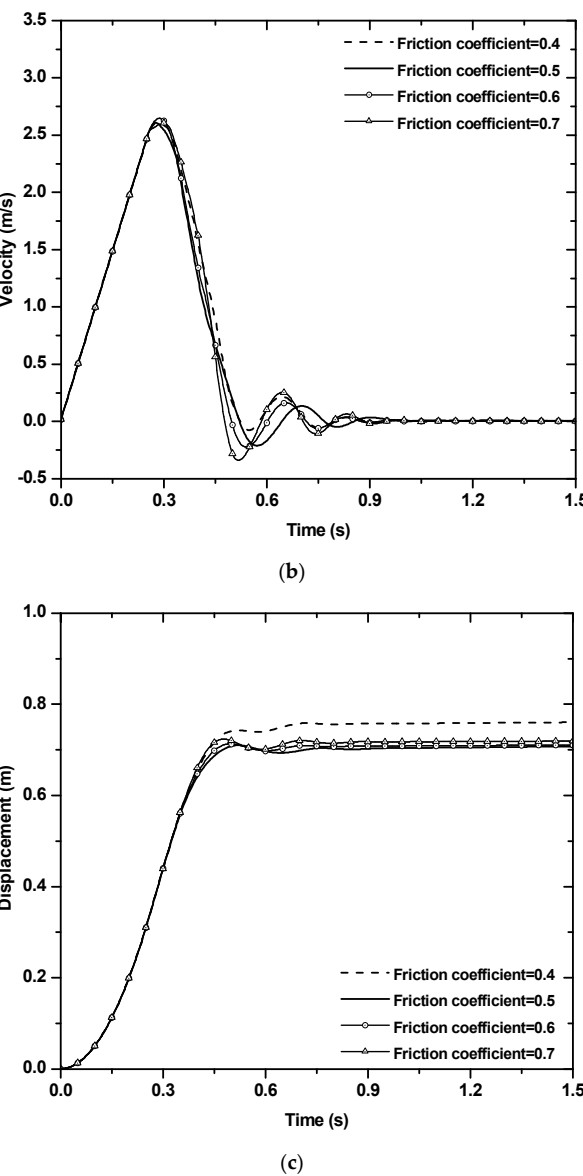

**Figure 14.** Vertical motion of weights clashing with gravel layers for different friction coefficients between the gravel and the weight (friction coefficient between gravel pieces = 0.5): (**a**) acceleration; (**b**) velocity; (**c**) displacement.

According to the simulation results, the friction between the gravel and the weight mainly affects the vertical motion of the penetrating weight. Although the fluctuation patterns of the acceleration are different for different friction coefficients, the vertical velocity and vertical displacement do not differ significantly. Specifically, the time-series vertical displacement and penetration depth appear to be affected by the friction coefficient when coefficients smaller than 0.5 are applied. This indicates that the dynamic behavior of the weight used for the drop test is mainly affected by the characteristics of the gravel layer. As shown in Figure 15, the friction-energy dissipation converges to a constant value as the considered friction coefficient increases. According to the simulation, the friction between the disk-type steel object and the gravel for the track ballast should be larger than 0.5.

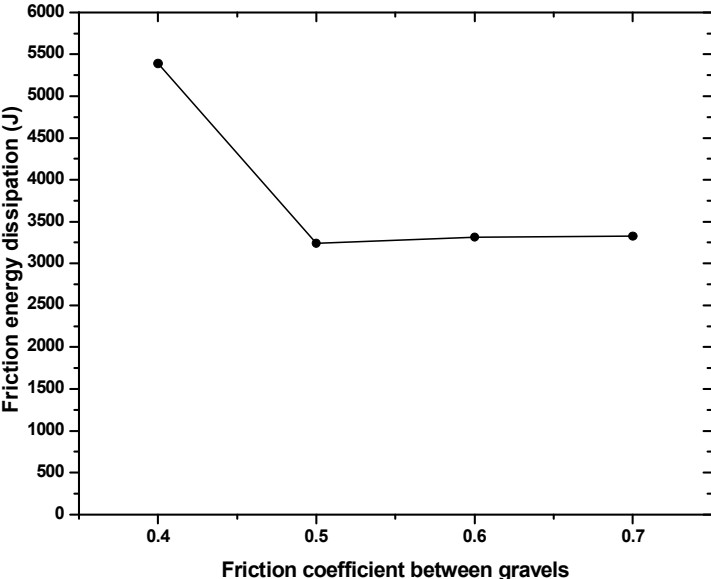

**Figure 15.** Comparison of friction-energy dissipation for different friction coefficients between the gravel and the weight.

## 5. Conclusions

In this study, a rational modeling method and an analysis for gravel-filled track ballast were investigated to simulate track-derailed train interaction. Because of the characteristics of the track ballast, a DEM-based modeling and analysis strategy was used for modeling the ballast. Once the track ballast was modeled with numerous rigid particles, the interaction between the track ballast and objects of any shape could be simulated.

- Based on the DEM approach, each piece of gravel is modeled as a rigid body to ensure an efficient simulation. To consider the elasticity of the gravel and the friction effects between the gravel pieces in contact, an appropriate contact model should be applied. In this study, the Hertz–Mindlin contact model was applied to consider the normal and tangential contact forces and the friction force. The nonlinear normal and tangential stiffnesses were estimated using the equation of the contact model considering the size, elastic modulus, and Poisson's ratio of the gravel. Apart from the contact between the pieces of gravel, a contact model between the gravel and contacted objects was also used along with the Hertz–Mindlin model.
- To validate the simulation method, a drop test was conducted, and the experiment was reproduced via a DEM-based simulation. The comparison results, including vertical acceleration, velocity, displacement, and the penetration depth of the freely dropped weights, exhibit very good agreement. Therefore, the feasibility and rationality of the simulation method was verified.
- The effects of the applied friction coefficient on the interaction between the gravel layer and the clashing object were studied. According to the results, the friction coefficient between the contacted gravel pieces significantly affects the geotechnical properties of the layer filled with gravel. The coefficient governs the interlocking effect in the gravel layer and, thus, this coefficient also affects the rigidity of the layer.
- The friction coefficient between the gravel and the object does not significantly affect the interaction between the layer and the clashing object. According to the simulation, the results converge when a coefficient over 0.5 is applied. In addition, based on both the experiment and simulation, a coefficient of 0.5 for the track ballast–steel wheel contact is appropriate.

Although the simulation results agree well with the test results, limitations still exist to be improved for simulation of derailed train-track ballast interaction as below:

- The effect of the shapes of different pieces of gravel could not be considered because simple rigid sphere clumps were used.
- To simulate derailed train-track ballast interaction, the collision cases which induce the shearing motion of the gravel of track as well as vertical motion should be further validated.
- For real track ballasts tamping effect should be taken into account.
- Further study is needed to find the rational range of friction coefficients between irregular gravels for real ballast track.

**Author Contributions:** Conceptualization, S.K., N.-H.L. and K.-J.K.; methodology, S.K. and H.-U.B.; software, S.K.; validation, S.K. and H.-U.B.; formal analysis, S.K. and H.-U.B.; investigation, S.K. and H.-U.B.; resources, N.-H.L.; data curation, K.-J.K. and H.U.B.; writing—original draft preparation, S.K. and N.-H.L.; writing—review and editing, S.K. and N.-H.L.; visualization, H.-U.B.; supervision, S.K.; project administration, N.-H.L.; funding acquisition, N.-H.L. All authors have read and agreed to the published version of the manuscript.

**Funding:** This research was funded by the Railway Technology Research Program, funded by the Ministry of Land, Infrastructure, and Transport of the Korean government, grant number 17RTRP-B122273-02.

**Conflicts of Interest:** The authors declare no conflict of interest.

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
