# Peer review of "DEM Analysis of Track Ballast for Track Ballast–Wheel Interaction Simulation"

_applsci, doi:10.3390/app10082717_

Round 1
Reviewer 1 Report
The introduction of the manuscript should be revisited. There are some reports that address the analytical treatment of soil-structure interaction [a-d].a. "Vibration analysis of a rigid circular disk embedded in a transversely isotropic solid." Journal of Engineering Mechanics 140.7 (2014): 04014048.
b. "Lateral translation of an inextensible circular membrane embedded in a transversely isotropic half-space." European Journal of Mechanics-A/Solids 39 (2013): 134-143.
c. "Rocking rotation of a rigid disk embedded in a transversely isotropic half-space." Civil Engineering Infrastructures Journal 47.1 (2014): 125-138.
d. "Finite element limit analysis of ultimate lateral pressure of XCC pile in undrained clay." Computers and Geotechnics 95 (2018): 240-246.
2. Font size in the figures should be bigger.
3. Terms used in Fig.4 should be defined in its caption.
4. Inset for some figures is recommended a zoomed in version of the plot to see the discrepancy of the results.
Author Response
Dear reviewer,
The authors appreciate the efforts of the editor and reviewers regarding our paper. The authors feel that reviewer provided many valid and useful comments for improvement of this paper. The authors prepared response to reviewer’s comments and revised the paper to deliver the significance of the presented work more clearly.
The following is point-by-point response to the reviewer’s comments:
Comment 1:
The introduction of the manuscript should be revisited. There are some reports that address the analytical treatment of soil-structure interaction [a-d].
- "Vibration analysis of a rigid circular disk embedded in a transversely isotropic solid." Journal of Engineering Mechanics 140.7 (2014): 04014048.
- "Lateral translation of an inextensible circular membrane embedded in a transversely isotropic half-space." European Journal of Mechanics-A/Solids 39 (2013): 134-143.
- "Rocking rotation of a rigid disk embedded in a transversely isotropic half-space." Civil Engineering Infrastructures Journal 47.1 (2014): 125-138.
- "Finite element limit analysis of ultimate lateral pressure of XCC pile in undrained clay." Computers and Geotechnics 95 (2018): 240-246.
Answer:
We have carefully reviewed all the suggested papers and added them to literature review section in the introduction chapter. (page 3 in the revised manuscript)
Comment 2:
Font size in the figures should be bigger.
Answer:
We checked all the figures and revised them. (Figs. 1, 4, 10, 11, 12, 13, 14, and 15)
Comment 3:
Terms used in Fig.4 should be defined in its caption.
Answer:
We added explanation of all the terms in Fig. 4. Please see the revised Fig. 4.
Comment 4:
Inset for some figures is recommended a zoomed in version of the plot to see the discrepancy of the results.
Answer:
To see the discrepancy of the results, we revised all the graphs (Figs. 10~15) with scale correction.
Reviewer 2 Report
A good aproach between the real conditions of the ballast bed and the DEM analisys.
Please mention if you have taken into account the ballast bed thickness under the sleeper and the geotechnical characteristics of the sub-bed layer.
Author Response
Dear reviewer,
The authors appreciate the efforts of the editor and reviewers regarding our paper.
In this study, the numerical method was suggested to simulate the track ballast-wheel interaction based on DEM approach. To verify the approach, we mainly conducted experiment and numerical simulation of dropped weight and gravel layer. The results show very good agreement, so we could conclude that the suggested modeling and simulation methods can be applied to analyze the derailed train and ballast interaction.
In this study, the details of the track components were not taken into account for the drop test. However, we used the gravel whose size is perfectly suitable for track ballasts based on Korean regulation.
Now, we are studying the derailed train-ballast interaction characteristics using this simulation approach. In this study, details of the track components are considered.
Reviewer 3 Report
There are several issues to this paper that need to be clarified.
- This article claimed to conduct “DEM Analysis of Track Ballast for Track Ballast-Wheel Interaction Simulation” from the title. However, other than the motivation discussed in the Introduction, the reviewer cannot find any discussion associated with “wheel” in the drop test or in the simulation. Perhaps the title should be renamed to better describe their work.
- The authors discussed the DCP system in the introduction. But how this system relates to their work, particularly for the drop test and simulation?
- In fact, the DCP system has not discussed clearly in the article, let along no reference, explanation and discussion for Fig. 1.
- On page 3, lines 66 to 68, the authors said “as mentioned earlier, the track ballast can be equivalently modeled based on the conventional FEM approach.” However, just in the previous paragraph, the authors criticize the use of FEM to simulate the track ballast. So which position the authors take? Approve the use of FEM or oppose it.
- On page 5, lines 177 to 181, the authors claimed they use Hertz contact model to determine the normal contact force. This model seems to be the crucial part of their model. The question is, there is no discussion about how this model is appropriate. Besides, no discussion about the determination of delta, which is the most important parameter changing with respect to time during the contact process.
- On page 9, lines from 251 to 253, the authors said Figs. 10 and 11 are “the motions of the weights, obtained from the drop test and the simulation.” Later on lines 255 to 259, they said “from the curves, we can select the time for the first contact between the dropped weight and the gravel layer, following which, the velocities and changes in displacement of the weight can be directly compared” and “as shown in the figures, the curves of the velocity and displacement of the weights obtained from the tests and the simulation are similar, including the magnitudes and the time required for full penetration.” So, are the figures 10—11 demonstrating the whole dropping test/simulation or just the contact process after the impact?
- 10(a) must be wrong. It doesn’t look like an acceleration.
- The results for the effects of friction coefficient on the friction energy dissipation against commonsense. We all know that the higher friction coefficient, the more friction force. Friction energy dissipation must therefore increase. The authors must carefully check their simulation, or provide more evidences to prove this is a physically correct phenomenon.
- The reference-list format is not consistent. It needs to be fixed.
Author Response
Dear reviewer,
The authors really appreciate the efforts of the reviewer regarding our paper. The authors feel that reviewer provided many valid and useful comments for improvement of this paper. The authors prepared response to reviewer’s comments and revised the paper to deliver the significance of the presented work more clearly.
*Please see the attachment

Reviewer 4 Report
Brief summary
The authors are aiming on a modelling approach for wheel-gravel interaction that could be used for modelling of train derailment events. They conduct a test and modelling combination as well as a combined FEM and DEM modelling approach. A drop test was chosen as a first rational event.
Broad comments
The manuscript is well written, but a more reflective discussion of the limitations of the method and the suitability of drop test to derailment events would be helpful. E.g. the free surface of gravel is orthogonal, and the acceleration of gravel is mainly a rebounding event, in contrast with a derailment event , when gravel can be more easily accelerated by a shearing motion and secondly pass through the free surface of gravel. The authors DEM model for gravel motion is based on a tangential frictional motion. The definition of a friction value for the spherical modelled irregular gravel bodies is stated to be decisive.
Specific comments
Abstract:
The abstract could be slighty adopted to a more reflective form, like:
This study aims to suggest a rational analysis method for the track ballast–wheel interaction that could be further developed to model the interaction in a train-derailment event, based on the discrete-element method (DEM).
Conclusions:
A formulation of an additional paragraph could be helpful to state what could or should be next steps to improve the testing and modelling approach to enable modelling of train derailment events.
Author Response
Dear reviewer,
The authors appreciate the efforts of the editor and reviewers regarding our paper. The authors feel that reviewer provided many valid and useful comments for improvement of this paper. The authors prepared response to reviewer’s comments and revised the paper reflecting all the comments and recommendation from you.
As you suggested, we clearly indicated the limitations of our study in the revised manuscript. As you mentioned, there are limitations which should be improved further to simulate derailed train-track ballast interaction, based on DEM approach. The limitations were summarized in the conclusion chapter, adding new paragraph. Please see the revised conclusion.
Also, we revised the first sentence of the abstract, as you suggested.
Thank you for your kind review.
Round 2
Reviewer 1 Report
My comments have been addressed in the revised manuscript. Therefore, the reviewer recommends the manuscript for publication.
Reviewer 3 Report
I have no more question about this revision.